# Reducing the Periplasmic Glutathione Content Makes *Escherichia coli* Resistant to Trimethoprim and Other Antimicrobial Drugs

Yajing Song,[a,b] Zhen Zhou,[c] Jing Gu,[a] Junmei Yang,[d] Jiaoyu Deng[a]

[a]CAS Key Laboratory of Special Pathogens and Biosafety, Wuhan Institute of Virology, Chinese Academy of Sciences, Wuhan, China
[b]University of Chinese Academy of Sciences, Beijing, China
[c]Hubei Key Laboratory of Environmental and Health Effects of Persistent Toxic Substances, Institute of Environment and Health, Jianghan University, Wuhan, China
[d]Zhengzhou Key Laboratory of Children's Infection and Immunity, Children's Hospital Affiliated with Zhengzhou University, Zhengzhou, China

**ABSTRACT** Although glutathione (GSH) has been shown to influence the antimicrobial effects of many kinds of antibiotics, little is known about its role in relation to trimethoprim (TMP), a widely used antifolate. In this study, several genes related to glutathione metabolism were deleted in different *Escherichia coli* strains (i.e., O157:H7 and ATCC 25922), and their effects on susceptibility to TMP were tested. The results showed that deleting *gshA*, *gshB*, *grxA*, and *cydD* caused TMP resistance, and deleting *cydD* also caused resistance to other drugs. Meanwhile, deleting *gshA*, *grxA*, and *cydD* resulted in a significant decrease of the periplasmic glutathione content. Supplementing exogenous GSH or further deleting glutathione importer genes (*gsiB* and *ggt*) restored TMP sensitivity to Δ*cydD*. Subsequently, the results of quantitative-reverse transcription PCR experiments showed that expression levels of *acrA*, *acrB*, and *tolC* were significantly upregulated in both Δ*grxA* and Δ*cydD*. Correspondingly, deleting *cydD* led to a decreased accumulation of TMP within bacterial cells, and further deleting *acrA*, *acrB*, or *tolC* restored TMP sensitivity to Δ*cydD*. Inactivation of CpxR and SoxS, two transcriptional factors that modulate the transcription of *acrAB-tolC*, restored TMP sensitivity to Δ*cydD*. Furthermore, mutations of *gshA*, *gshB*, *grxA*, *cydC*, and *cydD* are highly prevalent in *E. coli* clinical strains. Collectively, these data suggest that reducing the periplasmic glutathione content of *E. coli* leads to increased expression of *acrAB-tolC* with the involvement of CpxR and SoxS, ultimately causing drug resistance. To the best of our knowledge, this is the first report showing a linkage between periplasmic GSH and drug resistance in bacteria.

**IMPORTANCE** After being used extensively for decades, trimethoprim still remains one of the key accessible antimicrobials recommended by the World Health Organization. A better understanding of the mechanisms of resistance would be beneficial for the future utilization of this drug. It has been shown that the AcrAB-TolC efflux pump is associated with trimethoprim resistance in *E. coli* clinical strains. In this study, we show that *E. coli* can sense the periplasmic glutathione content with the involvement of the CpxAR two-component system. As a result, reducing the periplasmic glutathione content leads to increased expression of *acrA*, *acrB*, and *tolC* via CpxR and SoxS, causing resistance to antimicrobials, including trimethoprim. Meanwhile, mutations in the genes responsible for periplasmic glutathione content maintenance are highly prevalent in *E. coli* clinical isolates, indicating a potential correlation of the periplasmic glutathione content and clinical antimicrobial resistance, which merits further investigation.

**KEYWORDS** *Escherichia coli*, periplasm, glutathione, trimethoprim, resistance

Address correspondence to Jiaoyu Deng, dengjy@wh.iov.cn.

The biosynthesis of purines, thymine, glycine, methionine, and pantothenic acid requires folates as cofactors. Thus, folate is essential for all living organisms. Bacteria require *de novo* synthesis of folate, whereas mammals can uptake it from the environment. Therefore, the folate biosynthesis pathway is an ideal target for the design of new antimicrobial drugs (1). Trimethoprim (TMP), an antifolate that inhibits bacterial dihydrofolate reductase (DHFR), has been extensively used for decades as a broad-spectrum antibacterial drug in combination with sulfamethoxazole (SMX) (2, 3). However, the widespread use of TMP has caused increased antimicrobial resistance, similar to that occurring with other antimicrobial agents. Therefore, it has become urgent to further investigate the mechanisms of TMP resistance, which may facilitate the design of new antifolates.

Glutathione (GSH) is the major cellular thiol that protects cells against oxidative stress, and it is required for both disulfide-bond reduction and GSH-dependent peroxidase activities (4, 5). Recently, it has been proposed that GSH may contribute to antioxidant defense and redox homeostasis in the periplasm (6). In *Escherichia coli*, GSH is synthesized through two steps by γ-glutamyl cysteine synthetase (GshA) and GSH synthetase (GshB). Mutations in *gshA* and *gshB* are both devoid of GSH (7). Glutaredoxins (Grxs) are ubiquitous proteins that catalyze the reduction of disulfides via reduced GSH. *E. coli* contains three Grxs (Grx1, Grx2, and Grx3, encoded by *grxA*, *grxB*, and *grxC*, respectively) and two Grx-like proteins (Grx4, encoded by *grxD*, and NrdH). In *E. coli*, a continuous cycling of GSH between cells and the growth medium has been reported in exponentially growing aerobic cultures (6, 8); GsiABCD and GGT promote the uptake of GSH from the medium into the cytoplasm, and the CydDC transporter mediates the export of GSH from the cytoplasm to the periplasm.

Besides its role in maintaining cellular redox homeostasis, GSH can also influence the antimicrobial effects of different kinds of antibiotics, including beta-lactams, quinolones, and aminoglycosides (9–11). Exogenous GSH can reverse the effects of ciprofloxacin in *E. coli* by neutralizing ciprofloxacin-induced oxidative stress and increasing its efflux (9). Similarly, GSH-mediated protection against norfloxacin has been shown to be more efficient in the presence of the AcrAB-TolC efflux system (12). GSH-mediated protection has also been observed in aminoglycosides, such as spectinomycin. However, the decreased sensitivity of *E. coli* to streptomycin caused by exogenous GSH is not due to antioxidant-mediated scavenging of reactive oxygen species (11), and GSH-mediated increased antibiotic efflux is assumed to play a role in this process (13). These observations indicate the involvement of efflux systems in reversing the antimicrobial effects of antibiotics by GSH, which merits further investigation.

The AcrAB-TolC system is one of the best-characterized drug-efflux systems in *E. coli*, which belongs to the resistance-nodulation-cell division (RND) superfamily and has a wide range of substrates (14–16). In addition, SoxS has been shown to be a global transcriptional activator of *acrAB* and *tolC* (17, 18).

Although GSH has been shown to influence the antimicrobial effects of many drugs, whether it also affects that of TMP has remained unknown. To this aim, in the present study, several genes related to GSH biosynthesis, utilization, and transport were deleted in *E. coli* W3110, and their effects on TMP susceptibility were tested. Meanwhile, the effects of deleting these genes on GSH content in the bacterial periplasm were also determined. To further investigate the mechanism by which disrupting GSH metabolism affects TMP susceptibility, we measured the gene expression levels for *acrAB* and *tolC*, as well as the accumulation of TMP within bacterial cells upon drug treatment.

## RESULTS

**Deletion of *gshA*, *gshB*, *grxA*, and *cydD* causes resistance to multiple antimicrobial drugs in *E. coli*.** To elucidate the effects of GSH on TMP sensitivity in *E. coli*, genes involved in GSH biosynthesis (*gshA* and *gshB*), utilization (*grxA*, *grxB*, *grxC*, *grxD*, and *nrdH*), transport (*gsiA*, *gsiB*, *gsiC*, *gsiD*, *cydC*, *ggt*, and *cydD*), and recycling (*gor*) were knocked out in different strains of *E. coli* (W3110, MG1655, BW25113, O157:H7, and ATCC 25922). The results of drug susceptibility tests showed that four of the single-gene-deletion mutants (ΔgshA,

**TABLE 1** MICs of TMP for different strains

| Strain | MIC of TMP ($\mu$g/mL) |
| --- | --- |
| *E. coli* W3110 | 0.32 |
| *E. coli* W3110 Δ*gshA* | 1.28 |
| *E. coli* W3110 Δ*gshB* | 1.28 |
| *E. coli* W3110 Δ*grxA* | 1.28 |
| *E. coli* W3110 Δ*cydD* | 1.28 |
| *E. coli* W3110 Δ*cydD*Δ*grxA* | 1.28 |
| *E. coli* W3110 Δ*cydD*Δ*gshA* | 1.28 |
| *E. coli* W3110 Δ*cydD*Δ*gshB* | 1.28 |
| *E. coli* W3110 Δ*grxA*Δ*gshA* | 1.28 |
| *E. coli* W3110 Δ*grxA*Δ*gshB* | 1.28 |
| *E. coli* W3110 pCA24N | 0.32 |
| *E. coli* W3110 Δ*gshA* pCA24N | 1.28 |
| *E. coli* W3110 Δ*gshB* pCA24N | 1.28 |
| *E. coli* W3110 Δ*grxA* pCA24N | 1.28 |
| *E. coli* W3110 Δ*cydD* pCA24N | 1.28 |
| *E. coli* W3110 Δ*gshA* pCA24N::*gshA* | 0.16 |
| *E. coli* W3110 Δ*gshB* pCA24N::*gshB* | 0.32 |
| *E. coli* W3110 Δ*grxA* pCA24N::*grxA* | 0.32 |
| *E. coli* W3110 Δ*cydD* pCA24N::*cydD* | 0.32 |

Δ*gshB*, Δ*grxA*, and Δ*cydD*) were resistant to TMP (MIC = 1.28 mg/L) compared with their parental strain W3110 (MIC = 0.32 mg/L) (Table 1), suggesting that GSH affected TMP sensitivity in *E. coli*. Furthermore, we also tested the TMP sensitivity in the gene complement strains with the pCA24N vector, and the TMP MICs of the complement strains were similar to that of the wild-type strain (Table 1). The results of TMP (4 mg/L) exposure experiments also showed that the above four mutants were more resistant to TMP treatment (Fig. S1). In addition, deleting *gshA*, *gshB*, *grxA*, and *cydD* also caused TMP resistance in other *E. coli* strains (BW25113, MG1655, O157:H7, and ATCC 25922) (Table S1). We also found that double gene-knockout mutants—namely, Δ*cydD*Δ*grxA*, Δ*cydD*Δ*gshA*, Δ*cydD*Δ*gshB*, Δ*grxA*Δ*gshA*, and Δ*grxA*Δ*gshB*—showed the same levels of TMP resistance (MIC = 1.28 mg/L) as the four single-gene knockout mutants (Table 1), indicating that decreased GSH content in the periplasmic space might cause TMP resistance in *E. coli*. In addition, deleting *cydD* also caused resistance to other antimicrobial drugs, including aminoglycosides (e.g., kanamycin, neomycin, gentamicin, spectinomycin, and streptomycin), chloramphenicol, and rifampin. We also tested the susceptibilities to other antibiotics in the complemented strain of Δ*cydD*. The results showed that complementing the Δ*cydD* mutant with an intact *cydD* gene could completely restore the susceptibility to TMP, largely restore the susceptibility to kanamycin and neomycin, but only partially restore the susceptibility to gentamicin (Table S2). Susceptibility to chloramphenicol was not tested in the complemented strain, since the pCA24N vector contains a chloramphenicol resistance cassette.

**Mutations of *gshA*, *gshB*, *grxA*, *cydC*, and *cydD* are highly prevalent in *E. coli* clinical strains.** Our data showed that reducing the periplasmic GSH content through deleting *gshA*, *gshB*, *grxA*, *cydC*, or *cydD* made *E. coli* resistant to TMP. To further probe the clinical relevance of this observation, we analyzed a genome database of pathogenic *E. coli* downloaded from NCBI according to the research of Lopatkin et al. (19), searched for mutations in the coding region of *gshA*, *gshB*, *grxA*, *cydD*, and *cydC* (Table S3), and finally picked up the top 50 mutation sites to draw the graphics (Fig. 1). For example, the mutation *cydD*_p1679 (CGG, R->CAG, Q) was found in 2,137 of 7,992 pathogenic strains, and 1,195 of 3,903 were designated clinical strains ($P < 0.001$, Fisher's exact test) (Table S3). Overall, all of the five genes had many mutations in the pathogenic strains and showed a high proportion in the clinical strains, although the gene mutations may not necessarily mean the loss of gene function.

**Deletion of *gshA*, *grxA*, and *cydD* leads to decreased GSH content in the periplasm.** To further test the hypothesis that decreasing GSH content in the periplasmic space causes TMP resistance in *E. coli*, GSH levels in both the intracellular space and periplasm were compared among Δ*gshA*, Δ*grxA*, Δ*cydD*, and W3110. As expected,

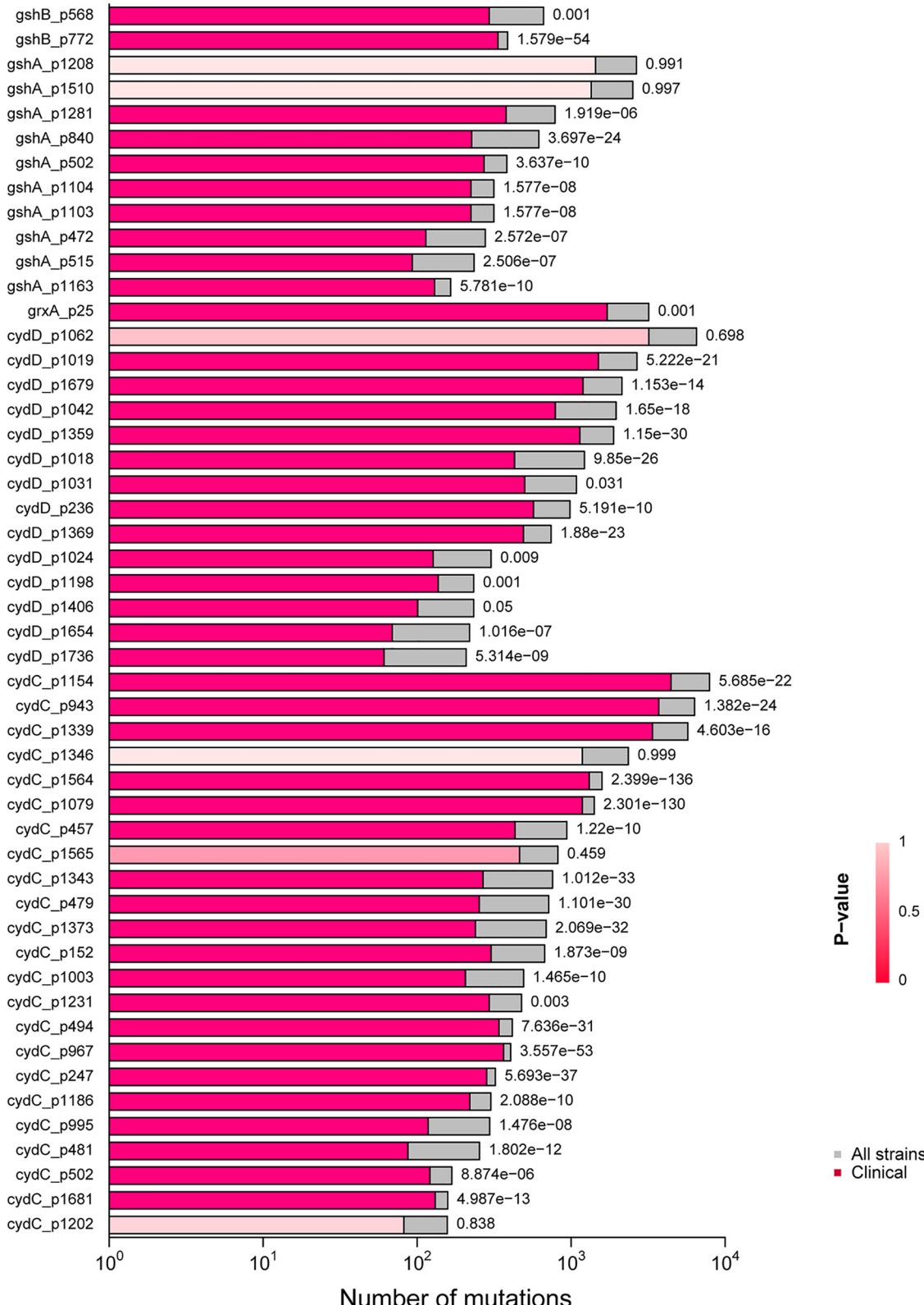

**FIG 1** Horizontal bar chart showing the high prevalence of mutations of *gshA*, *gshB*, *grxA*, *cydD*, and *cydC* in clinical *E. coli* strains. The coding mutations of *E. coli* *gshA*, *gshB*, *grxA*, *cydC*, and *cydD* were searched in a genome database of pathogenic *E. coli* strains downloaded from NCBI. The bars in shades of red indicate the number of strains with the specific mutation in clinical strains, and those in gray indicate the values for all strains. The *P* value of each mutation within the subset of clinical strains is indicated next to each bar; the reddish bars are shaded from dark to light red to indicate high significance to not significant, respectively. The *x* axis represents the number of mutations, and the *y* axis labels consist of the gene name and the position of the mutation.

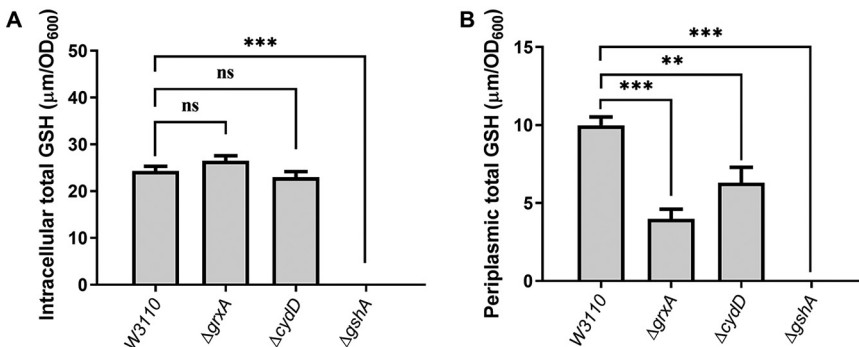

**FIG 2** GSH content in *E. coli* W3110, Δ*grxA*, Δ*cydD*, and Δ*gshA*. (A) Intracellular concentrations of total GSH. (B) Concentrations of total GSH in the periplasm. DTNB [5,5'-dithiobis-(2-nitrobenzoic acid)] assays were used to quantify GSH levels, which were normalized by the optical density at 600 nm ($OD_{600}$). The data represent the mean ± standard deviation (SD) from three independent experiments. *P* values were calculated using *t* tests (**, $P < 0.01$; ***, $P < 0.001$; ns, not significant).

deleting *gshA* led to significantly decreased GSH levels in both the intracellular and periplasmic spaces (Fig. 2A, Table S4). In contrast, deleting *grxA* and *cydD* only led to significantly decreased GSH levels in the periplasm (60% decrease for Δ*grxA* and 37% decrease for Δ*cydD*) (Fig. 2B, Table S4). In addition, the TMP-resistant phenotypes of Δ*gshA*, Δ*gshB*, Δ*grxA*, and Δ*cydD* were partially or completely reversed by the addition of exogenous GSH (Table S5). These data suggested that decreasing GSH content in the periplasm causes TMP resistance in *E. coli* W3110.

**Further inactivation of GSH importers restores TMP sensitivity to Δ*cydD*.** In *E. coli*, although CydDC is the only known transporter that exports GSH from the cytoplasm to the periplasm (20), several importers (e.g., GsiABCD and GGT) are involved in importing extracellular GSH into the cytoplasm to maintain the periplasmic concentration of GSH at an optimal level together with that of CydDC (6, 21). To further test the hypothesis that decreasing GSH content in the periplasm causes TMP resistance in *E. coli*, we knocked out *gsiA*, *gsiB*, *gsiAB*, *gsiC*, *gsiD*, and *ggt* in both W3110 and Δ*cydD*. The results of the subsequent TMP susceptibility tests showed that although the deletion of all these genes had no effects on TMP sensitivity in *E. coli* W3110, further deleting *gsiB* and *ggt* restored TMP sensitivity to Δ*cydD* (Table 2), demonstrating a connection between the periplasmic GSH content and TMP susceptibility.

**Deletion of *grxA* and *cydD* leads to increased expression of *acrAB*, *tolC*, and SoxS.** Since previous studies have indicated the involvement of the AcrAB-TolC efflux system in GSH-mediated protection against antibiotics (12), we investigated whether this system also played a role in TMP-resistant phenotypes caused by *grxA* and *cydD* gene deletions. The results of reverse transcription-quantitative PCR (RT-qPCR) experiments showed that the deletion of *grxA* and *cydD* both caused significantly increased

**TABLE 2** MICs of TMP for GSH efflux-related mutant strains

| Strain | MIC for TMP (μg/mL) |
|---|---|
| *E. coli* W3110 | 0.32 |
| *E. coli* W3110 Δ*gsiA* | 0.32 |
| *E. coli* W3110 Δ*gsiB* | 0.32 |
| *E. coli* W3110 Δ*gsiAB* | 0.32 |
| *E. coli* W3110 Δ*ggt* | 0.32 |
| *E. coli* W3110 Δ*gsiC* | 0.32 |
| *E. coli* W3110 Δ*gsiD* | 0.32 |
| *E. coli* W3110 Δ*cydD*Δ*gsiA* | 1.28 |
| *E. coli* W3110 Δ*cydD*Δ*gsiB* | 0.32 |
| *E. coli* W3110 Δ*cydD*Δ*gsiAB* | 0.32 |
| *E. coli* W3110 Δ*cydD*Δ*ggt* | 0.32 |
| *E. coli* W3110 Δ*cydD*Δ*gsiC* | 1.28 |
| *E. coli* W3110 Δ*cydD*Δ*gsiD* | 1.28 |

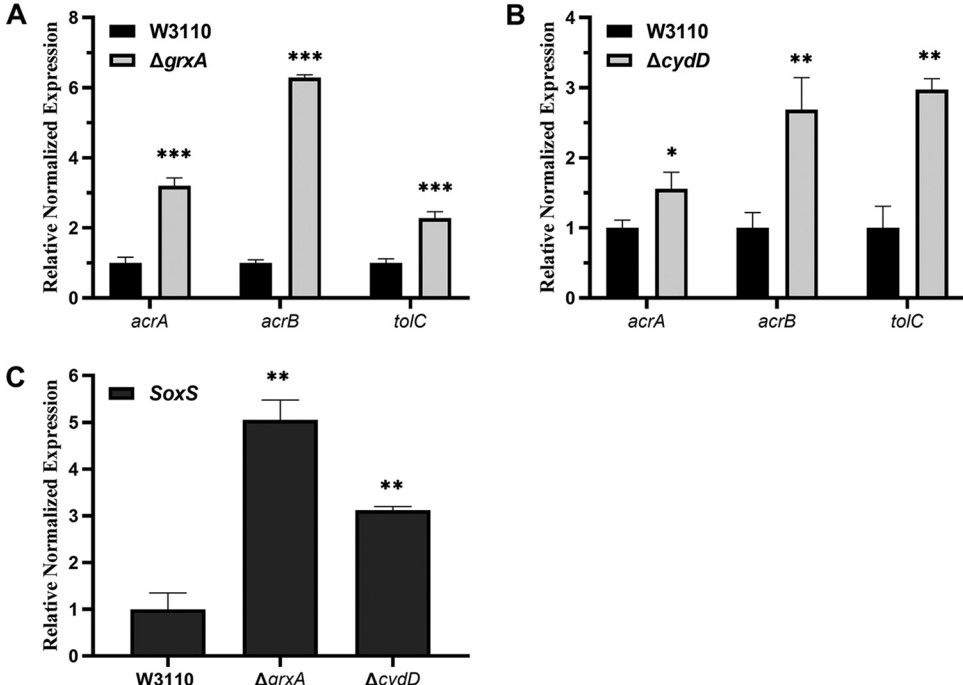

**FIG 3** Relative expression levels of *acrAB*, *tolC*, and *soxS* via RT-qPCR. (A) Expression levels of *acrAB* and *tolC* in *E. coli* W3110 and Δ*grxA*. (B) Expression levels of *acrAB* and *tolC* in *E. coli* W3110 and Δ*cydD*. (C) Expression levels of *soxS* in *E. coli* W3110, Δ*grxA*, and Δ*cydD*. The resulting cycle threshold ($C_T$) values were normalized using *gapA* as the reference gene. The data represent the mean ± SD from three independent experiments. *P* values were calculated using *t* tests (*, $P < 0.05$; **, $P < 0.01$; ***, $P < 0.001$).

expression levels of *acrA*, *acrB*, and *tolC* (Fig. 3). Specifically, 3.20-, 6.29-, and 2.28-fold increases in *acrA*, *acrB*, and *tolC* levels, respectively, were observed in the *grxA* deletion mutant (Fig. 3A, Table S6), and corresponding 1.55-, 2.69-, and 2.97-fold increases were observed in the *cydD* deletion mutant, compared with those in the wild-type strain W3110 (Fig. 3B, Table S6). Meanwhile, as SoxS is one of the transcriptional factors modulating the expression levels of *acrAB* and *tolC* (14, 17), we also found that the expression level of the *soxS* gene was also upregulated by 5.05- and 3.12-fold in the *grxA* and *cydD* deletion mutants, respectively (Fig. 3C, Table S6).

**Inactivating the AcrAB-TolC efflux system restores TMP sensitivity to Δ*grxA* and Δ*cydD*.** To further investigate the involvement of the AcrAB-TolC efflux system in TMP-resistant phenotypes caused by *grxA* and *cydD* gene deletions, the *acrA*, *acrB*, and *tolC* genes were knocked out in the wild-type strain W3110, as well as in Δ*grxA* and Δ*cydD*. The results of the subsequent TMP susceptibility tests revealed that deleting *acrA*, *acrB*, and *tolC* in the wild-type strain each caused increased sensitivity to TMP, and the further deletion of *acrA* completely reversed the TMP-resistant phenotypes of Δ*grxA* and Δ*cydD* (Table 3). The further deletion of *acrB* and *tolC* partially reversed the TMP-resistant phenotypes of Δ*grxA* and Δ*cydD*. These data suggest that the AcrAB-TolC efflux system was involved in the TMP-resistant phenotypes caused by *grxA* and *cydD* gene deletions.

**Deletion of *soxS* and *cpxR* restores TMP sensitivity to Δ*cydD*.** The above findings demonstrate that decreasing periplasmic GSH by deleting *grxA* and *cydD* led to increased expression levels of *acrA*, *acrB*, and *tolC*, which subsequently caused TMP resistance. To further investigate how changes in GSH content in the periplasm affect the transcription of *acrA*, *acrB*, and *tolC*—which usually occurs in the cytoplasm—two transcriptional-factor-coding genes, *soxS* and *cpxR*, were deleted in W3110 and Δ*cydD*. The results of drug susceptibility tests showed that although deleting *soxS* and *cpxR* only led to slightly increased sensitivity to TMP, these deletions fully restored TMP susceptibility to Δ*cydD* (Table 4), suggesting that SoxS and CpxR were involved in transferring GSH content from the periplasm

**TABLE 3** Effects of deletion mutants of efflux-pump genes on TMP sensitivity

| Strain | MIC for TMP ($\mu$g/mL) |
|---|---|
| E. coli W3110 | 0.32 |
| E. coli W3110 ΔcydD | 1.28 |
| E. coli W3110 ΔacrA | 0.08 |
| E. coli W3110 ΔacrB | 0.04 |
| E. coli W3110 ΔtolC | 0.16 |
| E. coli W3110 ΔcydDΔacrA | 0.08 |
| E. coli W3110 ΔcydDΔacrB | 0.16 |
| E. coli W3110 ΔcydDΔtolC | 0.16 |
| E. coli W3110 ΔgrxAΔacrA | 0.08 |
| E. coli W3110 ΔgrxAΔtolC | 0.16 |
| E. coli W3110 ΔgshAΔacrA | 0.08 |
| E. coli W3110 ΔgshAΔtolC | 0.32 |
| E. coli W3110 ΔgshBΔacrA | 0.08 |

into the cytoplasm. In addition, the further deletion of *cpxR* in Δ*cydD* caused significant decreases in the expression levels of *acrA*, *acrB*, *tolC*, and *soxS* (Fig. 4, Table S6).

**Deletion of *cydD* leads to decreased accumulation of TMP following TMP treatment.** Since the AcrAB-TolC efflux system has been shown to be involved in the TMP-resistant phenotype caused by *grxA* and *cydD* gene deletions, the accumulated levels of TMP between different strains were compared through liquid chromatography-mass spectrometry (LC-MS) analysis. As shown in Fig. 5 (Table S7), the deletion of *cydD* caused a significantly decreased accumulation of TMP within the bacterial cells, which was reversed by the further deletion of *acrA* and *tolC*. In contrast, deleting *acrA*, *acrB*, and *tolC* in W3110 each led to increased accumulation of TMP within the bacterial cells following TMP treatment.

## DISCUSSION

Many studies have shown that GSH influences the effects of different types of antimicrobial drugs. As one of the two thiol-dependent antioxidant systems in bacteria, the GSH system participates in the direct detoxification of reactive oxygen species, which has been suggested to play an important role in the bactericidal effects of many drugs, including TMP (9, 22). In contrast, GSH also seems to protect bacterial cells against some antibiotics with the involvement of drug efflux (12, 23). TMP, in combination with sulfamethoxazole (SMX), is one of the most commonly used antimicrobial drugs recommended by the World Health Organization. At present, the interplay between TMP and GSH has not been examined in bacteria. Here, we found that GSH influenced the antimicrobial effects of TMP, since disrupting its biosynthesis, utilization, and transport resulted in TMP resistance in *E. coli*.

Among the four genes related to GSH metabolism, *gshA* and *gshB* encode the two enzymes required for GSH biosynthesis in *E. coli*, *grxA* encodes one of the three glutaredoxins, and *cydD* encodes one subunit of the CydDC transporter, the latter of which is the only known transporter responsible for transporting GSH from the cytoplasm into the periplasm (20). Interestingly, the four mutants (Δ*gshA*, Δ*gshB*, Δ*grxA*, and Δ*cydD*) showed the

**TABLE 4** TMP MICs for the *cpxAR* and *soxS* gene mutants in Δ*cydD*

| Strain | MIC of TMP ($\mu$g/mL) |
|---|---|
| E. coli W3110 | 0.32 |
| E. coli W3110 ΔcydD | 1.28 |
| E. coli W3110 ΔcpxA | 2.56 |
| E. coli W3110 ΔcpxR | 0.16 |
| E. coli W3110 ΔcpxAΔcpxR | 0.32 |
| E. coli W3110 ΔsoxS | 0.16 |
| E. coli W3110 ΔcydDΔcpxA | 1.28 |
| E. coli W3110 ΔcydDΔcpxR | 0.32 |
| E. coli W3110 ΔcydDΔsoxS | 0.16 |

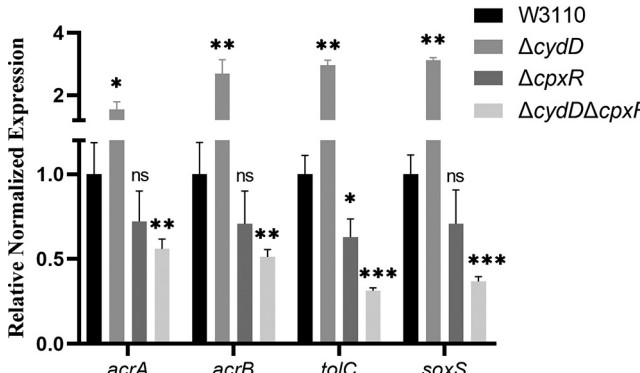

**FIG 4** Relative expression levels of *acrAB*, *tolC*, and *SoxS* in *E. coli* Δ*cpxR* and Δ*cydD*Δ*cpxR* via RT-qPCR. The resulting $C_T$ values were normalized using *gapA* as the reference gene. The data represent the mean ± SD from three independent experiments. *P* values were calculated using *t* tests (*, $P < 0.05$; **, $P < 0.01$; ***, $P < 0.001$; ns, not significant).

same levels of TMP resistance compared with that of their parental strain. This led us to speculate that all four mutants shared a common mechanism of TMP resistance, namely, that decreased GSH content in the periplasm causes TMP resistance. If this hypothesis is true, then any double-knockout mutant of the three genes (*gshA*, *grxA*, and *cydD*) should show the same level of TMP resistance as that of any corresponding single-gene knockout mutants, which our present results confirmed. Our hypothesis was further confirmed by the fact that the deletion of *gshA*, *grxA*, and *cydD* each led to a significant decrease in the periplasmic GSH content. Deleting *gshA* blocks the biosynthesis of GSH in *E. coli* and consequently decreases the periplasmic GSH content to a markedly low level. We found that deleting *grxA* caused a significant decrease in the periplasmic GSH content but only a slight increase in the intracellular GSH content. Although deleting *cydD* resulted in a significant decrease in the periplasmic GSH content, this decrease was less than that caused by deleting *gshA* and *grxA*. These data suggest the following in *E. coli*: (i) the majority of GSH in the periplasm is exported from the cytoplasm; (ii) CydDC is not the only exporter of GSH; and (iii) Grx1 modulates the export of GSH from the cytoplasm into the periplasm. Surprisingly, the deletion of *cydD* also caused a slight decrease in the intracellular GSH content. Previously, Holyoake et al. showed that deleting *cydD* in *E. coli* led to decreased expression of the CysB regulon (24), which is involved in the biosynthesis and transport of

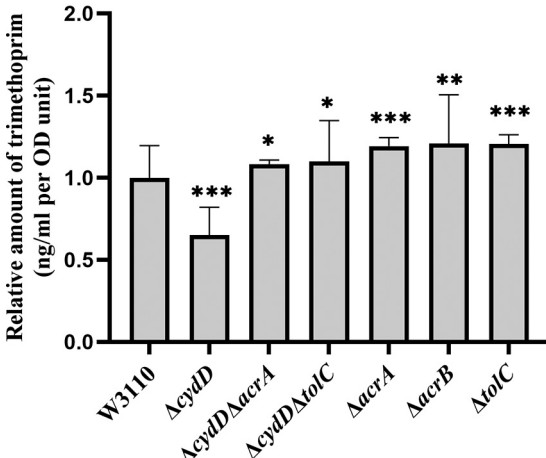

**FIG 5** Column chart representing the relative *E. coli* intracellular levels of TMP after challenging with 1 mg/L of the drug compared with the control sample W3110 detected by LC-MS analysis; the relative concentration levels of TMP were normalized to the $OD_{600}$ readings. Six replicates were used for each strain, and the data represent the mean ± SD from three independent experiments. *P* values were calculated using *t* tests (*, $P < 0.05$; **, $P < 0.01$; ***, $P < 0.001$).

cysteine, one of the precursors for GSH biosynthesis. We thus reasoned that the decreased production and uptake of cysteine caused by *cydD* deletion might lead to decreased biosynthesis of GSH.

Despite our present findings, it remains unclear as to how decreased GSH content in the periplasm leads to TMP resistance in *E. coli*. Previously, Jawali et al. showed that antibiotic-efflux machinery is involved in GSH-mediated decreased ciprofloxacin activity in *E. coli* (23). In our present study, through RT-qPCR analysis, we found that deleting *cydD* and *grxA* led to increased gene expression of *acrAB* and *tolC*. In addition, the further deletion of *acrA* completely reversed the TMP-resistant phenotypes of Δ*cydD* and Δ*grxA*, confirming that increased expression levels of *acrAB* and *tolC* caused TMP resistance in these two mutants. Increased expression levels of *acrAB* and *tolC* resulted in augmented efflux and diminished accumulation of TMP within the bacterial cells when treated with TMP, which is the direct cause of TMP resistance.

At present, it is unclear whether GSH affects the redox status of the periplasm, although some previous studies have provided related insights (6, 25). Although we found that decreased GSH content in the periplasm of *E. coli* led to increased expression of genes coding for the AcrAB-TolC drug-efflux pump, it remains unclear as to how bacteria transfer the information about fluctuations in the periplasmic GSH content into the cytoplasm, where transcription occurs. Fortunately, previous studies have already identified several transcriptional factors that modulate the expression of *acrAB* and *tolC*, including SoxS, one of the global regulators of multidrug resistance in *E. coli* (18, 26). Our RT-qPCR analysis data revealed that the gene expression levels of *soxS* were upregulated in the *grxA* and *cydD* deletion mutants compared to the wild-type strain, suggesting that elevated expression of SoxS activated the expression of *acrAB* and *tolC*. Additionally, the further deletion of *soxS* fully restored TMP sensitivity to the *cydD* deletion mutant, confirming the role of SoxS in mediating TMP resistance. However, there was no evidence showing the presence of SoxS in the periplasm or its shuttling between the cytoplasm and periplasm, indicating the involvement of other transcription factors that may interplay with GSH in the periplasmic space and thus affect the expression of *soxS*. CpxR is a response regulator in the CpxAR two-component system, which has been shown to sense and respond to the stress of periplasmic proteins misfolding and aggregating (27). Previously, Huang et al. showed that overexpression of CpxR in the *cpxR* deletion mutant of *Salmonella enterica* serovar Typhimurium resulted in the decreased expression of *soxS*, *acrB*, and *tolC* (28), indicating a role of CpxR in modulating the expression of *arcB* and *tolC* via SoxS. Our present data showed that the further deletion of *cpxR* led to significantly decreased expression of *soxS*, *acrA*, *acrB*, and *tolC*, thus restoring TMP sensitivity to the *cydD* deletion mutant and confirming the role of CpxR in mediating the TMP-resistant phenotype. Currently, there is no evidence showing that CpxR can directly modulate the expression of *acrA*, *acrB*, and *tolC* in *E. coli*. Based on our own present findings in combination with previous observations, we reasoned that decreasing the GSH content in the periplasm may activate the CpxAR two-component system, which would then upregulate the expression of *soxS*; increased expression of *soxS* would then cause increased expression of *acrAB* and *tolC*. Hence, the *E. coli* organism would then sense fluctuations in the GSH content in the periplasm and transduce this signal from the periplasm into the cytoplasm. However, future studies are required to elucidate how CpxR modulates the expression of *soxS*, *acrA*, *acrB*, and *tolC*.

In addition, we found that deleting *gshA*, *gshB*, *grxA*, or *cydD* also caused TMP resistance in pathogenic *E. coli* strains (O157:H7 and ATCC 25922). By analyzing a genome database of pathogenic *E. coli* downloaded from NCBI, we also found that mutations of these four genes are highly prevalent in *E. coli* clinical strains, indicating that mutations of these four genes may be associated with resistance to antimicrobial drugs, including TMP. Further studies are required to explore the physiological consequence of these mutations and to verify our hypothesis.

In summary, we found that decreasing the periplasmic GSH content in *E. coli* led to the increased expression of *soxS* via CpxR, which then activated the expression of genes

coding for the AcrAB-TolC efflux pump, ultimately causing TMP resistance. Thus, we identified a novel interplay between periplasmic GSH and drug susceptibility in *E. coli*. In addition, our findings provide the first evidence that signals reflecting fluctuations in periplasmic GSH content in *E. coli* can be transduced from the periplasmic space into the cytoplasm with the involvement of SoxS and CpxAR, which merits further investigation.

## MATERIALS AND METHODS

**Construction of bacterial strains and culture conditions.** The strains used in the present study were derivatives of *E. coli* K-12 W3110, MG1655, and BW25113, which were obtained from laboratory stock. The enterohemorrhagic *E. coli* (EHEC) strain O157:H7 was purchased from the China Center of Industrial Culture Collection (CICC) and numbered CICC 21530, and the standard strain ATCC 25922 was obtained from Qingdao Rishui Bio-Technologies Co., Ltd. All strains were grown in LB medium. The gene knockout mutants were constructed using the $\lambda$-red recombination system, as previously described (29). The mutants were verified by junction PCR and subsequent sequencing using primers that anneal to the genomic region outside the recombination locus. The double knockout mutants were generated using the same procedures, except that the kanamycin-resistance gene of the single-gene knockout mutant was first eliminated by transforming the plasmid pCP20. The gene-complemented strains of all of the mutants were constructed as follows: the target gene was amplified and cloned into the pCA24N vector and then transformed into the corresponding mutant strain. All plasmids, primers, and strains that we used are listed in Table S9.

**Preparation of reagents and drug susceptibility tests.** The reagents used in this study, including TMP and GSH, were purchased from Sigma-Aldrich. All reagents were solubilized according to the manufacturer's recommendations, and stock solutions were filter-sterilized and stored in aliquots at $-20°C$.

For drug susceptibility tests, bacterial cells were grown to an optical density at 600 nm ($OD_{600}$) of 0.8 and were diluted to approximately $10^5$ CFU/mL via 10-fold serial dilutions in fresh LB. Then, 10-fold serial dilutions were plated onto solid LB agar plates containing various concentrations of different tested drugs. The plates were incubated overnight at 37°C. The MIC was defined as the lowest concentration of a given compound required to inhibit the growth of 99% of bacterial CFU.

**Measurement of the growth curve and the time-kill curve.** Bacterial cells were grown to the mid-log phase ($OD_{600}$, ~0.8) and diluted to an $OD_{600}$ of 0.1 in fresh LB medium; then, approximately $10^6$ CFU/mL was set as the initial bacteria amount. Bacterial cells were grown aerobically at 37°C. For the bacterial growth curve, the $OD_{600}$ was continuously observed using a Synergy H1 hybrid reader (BioTek, USA) with shaking at 150 rpm. For the TMP (4 mg/L) exposure experiment, static cultivation serial dilutions were performed before plating onto LB plates.

**Comparative analysis of target gene mutations in an NCBI genome database of pathogenic *E. coli*.** Following the method of Lopatkin et al., we downloaded a genome database of pathogenic *E. coli* from NCBI, which contains 10,906 genomes classified as clinical, environmental, or other (19). We then mapped the target coding sequence of *E. coli* W3110 *gshA*, *gshB*, *grxA*, *cydC*, and *cydD* using the alignment software minimap2 to extract variation information. Specifically, we used the python module pysam to extract the base information of each genome for each gene locus, respectively, and to retain single-nucleotide polymorphism (SNP) and indel information. Then, comparative analysis of the variation information associated with the phenotype and statistics of the proportion of mutations in the clinical strains were performed using Fisher's test with the R programming language. We filtered out the nonsense mutations in the process and finally picked up the top 50 mutation sites to draw graphics.

**Periplasm separation and GSH measurements.** Periplasmic fractions were isolated using a modified osmotic-shock procedure (20). In brief, bacterial cells were grown to the mid-log phase ($OD_{600}$, ~0.8), harvested by centrifugation (3,000 × *g* for 10 min at 4°C), resuspended in 1 mL supernatant solution, and then supplemented with an equal volume of 20% (wt/vol) sucrose solution containing 10 mM Tris-HCl (pH 8.0). Subsequently, lysozyme (final concentration, 50 $\mu$g/mL) and EDTA (final concentration, 1 mM) were added. After incubation on ice for 30 min, centrifugation (10,000 × *g* for 5 min at 4°C) was performed, and then the pellets were collected and resuspended in 10 mM Tris-HCl (pH 8.0). After incubation on ice for 30 min, centrifugation (12,000 × *g* for 10 min at 4°C) yielded a supernatant fraction that was retained as the periplasmic fraction.

The periplasmic GSH levels were measured using a total GSH assay kit (Beyotime Biotech) according to the manufacturer's instructions. The periplasmic-fraction supernatant was treated with DTNB [5,5′-dithiobis-(2-nitrobenzoic acid)] in combination with GSH reductase enzyme and NADPH. Finally, the absorbance values were measured at a wavelength of 412 nm, using a Synergy H1 hybrid reader (BioTek). The GSH levels were quantified against a corresponding standard curve.

**RT-qPCR.** The expression levels of *acrA*, *acrB*, and *tolC* were compared between the wild-type strain and mutant strains via RT-qPCR. All bacterial cells were grown in LB broth to the mid-log phase ($OD_{600}$, ~0.8). Total RNA was isolated using an RNeasy kit (Qiagen, Germany), and cDNA was synthesized with a ReverTra Ace qPCR kit (Toyobo) according to the manufacturer's instructions. Quantification of gene expression levels was performed using real-time qPCR analysis on a 7900HT sequence-detection system (ABI, USA) with ABI Power SYBR green PCR master mix. The primers were designed and synthesized by the Shanghai Biotechnology Corporation. We used the comparative cycle threshold ($C_T$) value method for RT-PCR in this study; the resulting $C_T$ values were normalized using *gapA* as the reference gene, and the final relative gene expression level was normalized using the wild-type strain (23). Detailed information about the primers is given in Table S9 of the supplemental material.

**Quantification of intracellular TMP levels in *E. coli* via liquid chromatography-mass spectrometry.** The bacterial intracellular TMP levels were estimated using liquid chromatography-mass spectrometry (LC-MS),

and intracellular TMP quantification from *E. coli* was performed as previously described (30). In brief, all tested *E. coli* strains were grown in LB broth to the mid-log phase (OD600 ~0.8), washed once with fresh LB broth, diluted to an $OD_{600}$ of approximately 0.2, and then challenged with 1 mg/L of TMP. After incubation at 37℃ on a shaker (200 rpm) for 30 min, 10 mL of each culture was collected for metabolic quenching. For each sample, a double volume of 60% cold methanol was added, and cellular pellets were collected by centrifugation (1800 g for 10 min) for the subsequent extraction of metabolites. The cellular pellets were resuspended in 1 mL 80% cold methanol, and three rounds of freeze-thawing were performed to maximize the extraction of intracellular metabolites. The samples extracted from intracellular fractions were then subjected to LC-MS analysis, and the TMP reference standard was obtained against the appropriate concentration level. The details of these methods are provided in Table S8 and in the supplemental material for LC-MS. The TMP levels within the intracellular extracts were inferred against the standard curve, and the intracellular concentration levels were normalized to the $OD_{600}$ readings taken from the original bacterial cultures.

## SUPPLEMENTAL MATERIAL

Supplemental material is available online only.
**SUPPLEMENTAL FILE 1**, PDF file, 0.3 MB.
**SUPPLEMENTAL FILE 2**, XLSX file, 0.03 MB.

## ACKNOWLEDGMENTS

This work was supported by the Strategic Priority Research Program of the Chinese Academy of Sciences (grant number XDB29020000).
We thank LetPub for linguistic assistance during the preparation of the manuscript.

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
