## [Reviewer comments · Microbiology Spectrum]

Microbiology Spectrum

Reducing the Periplasmic Glutathione Content Makes *Escherichia coli* Resistant to Trimethoprim and Other Antimicrobial Drugs

Yajing Song, Zhen Zhou, Jing Gu, Junmei Yang, and Jiaoyu Deng

Corresponding Author(s): Jiaoyu Deng, Key Laboratory of Special Pathogens and Biosafety, Wuhan Institute of Virology, Chinese Academy of Sciences

Review Timeline:

Submission Date:	June 30, 2021
Editorial Decision:	August 14, 2021
Revision Received:	October 8, 2021
Accepted:	October 30, 2021

Editor: Hui Wang

Reviewer(s): The reviewers have opted to remain anonymous.

Transaction Report:

DOI: <https://doi.org/10.1128/Spectrum.00743-21>

August 14, 2021

Prof. Jiaoyu Deng
Key Laboratory of Special Pathogens and Biosafety, Wuhan Institute of Virology, Chinese Academy of Sciences
middle street of xiaohongshan, NO.44
Wuhan, Hubei 430071
China

Re: Spectrum00743-21 (Reducing the Periplasmic Glutathione Content Makes Escherichia coli Resistant to Trimethoprim and Other Antimicrobial Drugs)

Dear Prof. Jiaoyu Deng:

Thank you for submitting your manuscript to Microbiology Spectrum. When submitting the revised version of your paper, please provide (1) point-by-point responses to the issues raised by the reviewers as file type "Response to Reviewers," not in your cover letter, and (2) a PDF file that indicates the changes from the original submission (by highlighting or underlining the changes) as file type "Marked Up Manuscript - For Review Only". Please use this link to submit your revised manuscript - we strongly recommend that you submit your paper within the next 60 days or reach out to me. Detailed information on submitting your revised paper are below.

Link Not Available

Sincerely,

Hui Wang

Journals Department
Reviewer comments:

Reviewer #1 (Comments for the Author):

This manuscript describes the relationship between glutathione (GSH) and Trimethoprim (TMP) and confirmed GSH affects the antimicrobial effects of TMP. The results showed that decreasing periplasmic GSH content in *E. coli* led to increased expression of *soxS* via CpxR, which then activated the expression of genes coding for the AcrAB-TolC efflux pump, ultimately causing TMP resistance. Unfortunately, the more detailed and solid experiment evidence regarding how this process happened was unavailable in current version. Therefore, its novelty is relatively limited. Furthermore, there are some problems with the experiment methods which should be modified extensively. Here below are my specific major comments.

1 The authors need to notice that the MIC value of TMP resistance of Enterobacteriaceae should be greater than or equal to 16 µg/mL according to CLSI 2020. The standard strain ATCC 25922 was not used as a reference for the susceptibility testing. The agar used in the drug sensitivity test should be MHA, according to the standard methods given by CLSI. Therefore, The statement of TMP resistance (MIC=1.28 µg/mL) in current study needs to be further verified.

2 The standard curve method or comparative Ct value method for RT-PCR was not mentioned.

- 3 The authors claimed that after knocking out the *cydD* gene, the susceptibility to other antibiotics is reduced. Does that determine the sensitivity of other antibiotics in gene complement strains?
- 4 Efflux pump is a general antimicrobial mechanism against many kinds of antibiotics. So, the resistance of drugs that are the substrate of this mechanism would be all influenced by interrupting the GSH metabolism?
- 5 In the analysis of the mutations of genes involved in GSH metabolism of the clinical *E. coli* strain, it should be emphasized that mutations may not necessarily mean the loss the function of the gene.

Reviewer #2 (Comments for the Author):

Deng and colleagues reported the reduced accumulation of periplasmic GSH may lead to antibiotic resistance by up-regulating several resistance genes such as *acrAB-tolC*. This work updates the understanding towards how GSH metabolism affects the antimicrobial resistance of bacteria and sheds the light on the potential association between bacterial metabolic progress and their non-metabolic phenotypes. The experiments were well designed and the manuscript is of interest to the researchers in the microbiology field. However, there are some major concerns to be addressed.

Major:

1. The authors provided convincing but only phenotypical evidence to elucidate the linkage between periplasmic GSH and drug resistance. How will be the potent machinery of bacterial GSH enhance the antibiotic tolerance via the SOS-efflux pathway? As we know *soxS* is highly-related to the intracellular ROS level, which generally quenched by GSH. Will the ROS be a checkpoint between periplasmic GSH and upregulated expression of efflux? The authors better perform experiments to see whether ROS involved in this mechanism.
2. The authors indicate that it is the periplasmic not cytoplasmic GSH functionalized the antibiotic resistance. It will be of great significance to decipher this site-dependent effect of GSH.
3. I noticed that there are the slight differences in resistant phenotypes among mutant from different *E. coli* strains. Could author explain why choose W3100 as the model strain throughout the manuscript.
4. The manuscript was not given in proper scientific English and should be tidied up prior to resubmission.

Minor:

Line 54: 'living creatures?'

Line 61: occurring to

Line 67-68: synthesized through 2-step reaction

Line 77: besides its role

Line 144: 'behaved'?

Line 125-136: this section appears with weak consistency with other parts, and associated discussion may be necessary

Line 195: what does 'slight' stand for? Is it of statistical significance or just a numerical decrease?

Staff Comments:

Preparing Revision Guidelines

For complete guidelines on revision requirements, please see the Instructions to Authors at [link to page]. **Submissions of a paper that does not conform to Microbiology Spectrum guidelines will delay acceptance of your manuscript.**

Please return the manuscript within 60 days; if you cannot complete the modification within this time period, please contact me. If you do not wish to modify the manuscript and prefer to submit it to another journal, please notify me of your decision immediately so that the manuscript may be formally withdrawn from consideration by Microbiology Spectrum.

If you would like to submit an image for consideration as the Featured Image for an issue, please contact Spectrum staff.

Reducing the Periplasmic Glutathione Content Makes Escherichia coli Resistant to Trimethoprim and Other Antimicrobial Drugs

Major concerns:

Deng and colleagues reported the reduced accumulation of periplasmic GSH may lead to the antibiotic resistance by up-regulating several resistance genes such as *acrAB-tolC*. This work updates the understanding towards how GSH metabolism affects the antimicrobial resistance of bacteria and sheds the light on the potential association between bacterial metabolic progress and their non-metabolic phenotypes. The experiments were well designed and the manuscript is of interests to the researchers in microbiology field. However, there are some major concerns to be addressed before consideration for acceptance.

Major:

1. The authors provided convincing but only phenotypical evidences to elucidate the linkage between periplasmic GSH and drug resistance. How will be the potent machinery of bacterial GSH enhance the antibiotic tolerance via SOS-efflux pathway? As we know *soxS* is highly-related to the intracellular ROS level, which generally quenched by GSH. Will the ROS be a checkpoint between periplasmic GSH and upregulated expression of efflux? The authors better perform experiments to see whether ROS involved in this mechanism.
2. The authors indicate that it is the periplasmic not cytoplasmic GSH functionalized the antibiotic resistance. It will be of great significance to decipher this site-dependent effect of GSH.
3. I noticed that there are the slight differences in resistant phenotypes among mutant from different *E. coli* strains. Could author explain why choose W3100 as the model strain throughout the manuscript.
4. The manuscript was not given in proper scientific English and should be tidied up prior to resubmission/publication.

Minor:

Line 54: 'living creatures?'

Line 61: occurring to

Line 67-68: synthesized through 2-step reaction

Line 77: besides its role

Line 144: 'behaved'?

Line 125-136: this section appears with weak consistency with other parts, and associated discussion may be necessary

Line 195: what does 'slight' stand for? Is it of statistical significance or just numerical decrease?

Detailed response to reviewers

To Reviewer # 1

(1) Original comment: The authors need to notice that the MIC value of TMP resistance of Enterobacteriaceae should be greater than or equal to 16 µg/mL according to CLSI 2020. The standard strain ATCC 25922 was not used as a reference for the susceptibility testing. The agar used in the drug sensitivity test should be MHA, according to the standard methods given by CLSI. Therefore, the statement of TMP resistance (MIC=1.28 µg/mL) in current study needs to be further verified.

Reasons and modifications: Thank you for your comments. We obtained the standard strain ATCC 25922 and also the MHA medium. Unfortunately, we noticed that the instruction for performing the TMP susceptibility using the MHA medium (CLSI2020) says that the medium may contain antagonists of the drug. Our data also showed that the TMP MIC of *E. coli* is much higher in the MHA medium than that in the LB medium (Table 1). Therefore, we constructed four single gene deletion mutants based on ATCC 25922 ($\Delta gshA$, $\Delta gshB$, $\Delta grxA$, and $\Delta cydD$), and found that all those four mutants were more resistant to TMP (Table 2).

Table 1 Susceptibility of *E. coli* to TMP in different growth medium.

Strains	MIC for TMP ($\mu\text{g/mL}$)	
	LB medium	MHA medium
E. coli W3110	0.32	2.56
E. coli BW25113	0.32	>32
E. coli MG1655	0.32	>32
EHEC O157:H7	0.64	>32
E. coli ATCC 25922	2.56	>32

Table 2 MICs of different strains derived from *E. coli* ATCC 25922 to TMP (in LB medium).

Strains	MIC for TMP ($\mu\text{g/mL}$)
E. coli ATCC 25922	2.56
E. coli ATCC 25922 ΔgshA	20.48
E. coli ATCC 25922 ΔgshB	10.24
E. coli ATCC 25922 ΔgrxA	5.12
E. coli ATCC 25922 ΔcydD	10.24

(2) Original comment: The standard curve method or comparative Ct value method for RT-PCR was not mentioned.

Reasons and modifications: Thank you for the reminder. The manuscript has been revised according to your suggestion (line 279 of the revised MS).

(3) Original comment: The authors claimed that after knocking out the *cydD* gene, the susceptibility to other antibiotics is reduced. Does that determine the sensitivity of other antibiotics in gene complement strains?

Reasons and modifications: Thank you for your suggestions. The manuscript has

been revised according to your suggestions (lines 124–129 of the revised MS). We also tested the sensitivities to other antibiotics of the complemented strain along with the $\Delta cydD$ mutant. The results showed that (Table S2 of the revised manuscript), complementing $\Delta cydD$ with an intact *cydD* gene could completely restore the susceptibility to TMP, largely restore the susceptibility to kanamycin and neomycin, but only partially restore the susceptibility to gentamycin. Susceptibility to chloramphenicol, the complemented strain was not tested since the pCA24N vector contains a chloramphenicol resistance cassette (Table S2).

TABLE S2 Susceptibility of *E. coli* W3110 $\Delta cydD$ and complement strain to multiple antimicrobial drugs with different IPTG concentration.

Strains	IPTG (μm)	MIC ^a ($\mu\text{g/mL}$) of							
		TMP	K	Neo	Gen	Spe	SM	Cm	RIF
E. coli W3110		0.32	5	5	2	20	8	5	8
E. coli W3110 $\Delta cydD$	0	1.28	40	40	16	40	32	10	16
E. coli W3110 $\Delta cydD$ pCA24N:: cydD		0.32	10	10	8	40	16	—	16
E. coli W3110		0.32	5	5	2	20	8	5	8
E. coli W3110 $\Delta cydD$	2.5	1.28	40	40	16	40	32	10	16
E. coli W3110 $\Delta cydD$ pCA24N:: cydD		0.32	10	10	8	40	16	—	16
E. coli W3110		0.32	5	5	2	20	8	5	8
E. coli W3110 $\Delta cydD$	5	1.28	40	40	16	40	32	10	16
E. coli W3110 $\Delta cydD$ pCA24N:: cydD		0.32	20	20	8	40	16	—	16

^aTMP, trimethoprim; K, kanamycin; Neo, neomycin; Gen, gentamicin; Spe, spectinomycin; SM, streptomycin; Cm, chloramphenicol; RIF, rifampin.

(4) Original comment: Efflux pump is a general antimicrobial mechanism against many kinds of antibiotics. So, the resistance of drugs that are the substrate of this mechanism would be all influenced by interrupting the GSH metabolism?

Reasons and modifications: Thank you for your suggestion. Based on the obtained data, it seems that resistance of drugs that are substrates of the AcrAB-TolC efflux pump would be all affected when the periplasmic content of GSH in *E. coli* is decreased. But GSH also play an important role in maintaining redox state of the cytoplasm, and this also needs to be considered.

(5) Original comment: In the analysis of the mutations of genes involved in GSH metabolism of the clinical *E. coli* strain, it should be emphasized that mutations may not necessarily mean the loss the function of the gene.

Reasons and modifications: Thank you for your suggestion. The manuscript has been revised according to your suggestion (line 141 of the revised MS).

To Reviewer # 2

(1) Original comment: The authors provided convincing but only phenotypical evidence to elucidate the linkage between periplasmic GSH and drug resistance. How will be the potent machinery of bacterial GSH enhance the antibiotic tolerance via the SOS-efflux pathway? As we know *soxS* is highly-related to the intracellular ROS level, which generally quenched by GSH. Will the ROS be a checkpoint between periplasmic GSH and upregulated expression of efflux? The authors better perform experiments to see whether ROS involved in this mechanism.

Reasons and modifications: Thank you for your suggestions. We measured intracellular ROS levels of different strains (W3110, W3110 Δ *cydD*, W3110 Δ *soxS*, W3110 Δ *cydD* Δ *soxS*) using the probe H₂DCFDA (Beyotime Biotechnology, China) as previously described (1). In brief, bacterial cells were grown with an initial OD₆₀₀ of 0.1 at 37 °C in LB medium to mid-log phase (OD₆₀₀ ~0.8) , harvested by centrifugation (3,000 g for 10 min at 4°C), washed twice with phosphate buffered saline (PBS, pH 7.0), resuspended in 1 mL 10 μm/L H₂DCFDA solution (dissolved in PBS buffer). After incubation at 37°C for 30 min, bacterial cells were harvested by centrifugation (3,000 g for 10 min), washed twice with PBS buffer, then resuspended in 0.5 mL PBS buffer. After that, the cell density (OD₆₀₀) was determined, and the fluorescence intensity was measured by a SynergyH1 Hybrid reader (BioTek, USA) (excitation, 488 nm; emission, 525 nm). The results showed that although the intracellular ROS level significantly increased upon TMP treatment, no statistical difference could be observed between the WT and Δ *cydD*, as shown in the below

figure.

FIG. The Intracellular ROS level of *E. coli* W3110, $\Delta cydD$, $\Delta soxS$, and $\Delta cydD\Delta soxS$ upon TMP treatment. The intracellular ROS level was normalized by the OD_{600} . Data represent the mean \pm standard deviation (SD) from three independent experiments. *P* values were calculated using *t*-tests (*, $P < 0.05$; **, $P < 0.01$; ***, $P < 0.001$).

1. Wang J, Zhang Y, Chen Y, Lin M, Lin Z. 2012. Global regulator engineering significantly improved *Escherichia coli* tolerances toward inhibitors of lignocellulosic hydrolysates. *Biotechnol Bioeng* 109:3133-42.

(2) Original comment: The authors indicate that it is the periplasmic not cytoplasmic GSH functionalized the antibiotic resistance. It will be of great significance to decipher this site-dependent effect of GSH.

Reasons and modifications: Thank you for your suggestion. Our data showed that decreasing the periplasmic GSH content leads to increased expression of the AcrAB-TolC efflux pump encoding genes, making *E. coli* more resistant to multiple antimicrobial drugs including TMP. However, the mechanism with which *E. coli* senses the fluctuation of the periplasmic GSH content still needs further investigation. It seems that the CpxAR system may play a key role, but further verification is

certainly required.

(3) Original comment: I noticed that there are the slight differences in resistant phenotypes among mutant from different *E. coli* strains. Could author explain why choose W3100 as the model strain throughout the manuscript.

Reasons and modifications: Thank you for your suggestions. We agree that there are slight differences in resistance phenotype among mutants from different *E. coli* strains. We used as many different strains as possible to avoid the possibility of strain bias. According to the suggestions of another reviewer, we also tried another strain, *E. coli* ATCC 25922, and the results were shown in the below table. W3110 is one of the most commonly used K12 strains in research labs, which was a generous gift from Dr. Ying Zhang (2).

MICs of different strains derived from *E. coli* ATCC 25922 to TMP.

Strains	MIC for TMP ($\mu\text{g/mL}$)
E. coli ATCC 25922	2.56
E. coli ATCC 25922 ΔgshA	20.48
E. coli ATCC 25922 ΔgshB	10.24
E. coli ATCC 25922 ΔgrxA	5.12
E. coli ATCC 25922 ΔcydD	10.24

- Li Y, Zhang Y. 2007. PhoU is a persistence switch involved in persister formation and tolerance to multiple antibiotics and stresses in *Escherichia coli*. *Antimicrob Agents Chemother* 51:2092-9.

(4) Original comment: The manuscript was not given in proper scientific English and should be tidied up prior to resubmission.

Reasons and modifications: Thank you for your suggestions. The revised manuscript has been reviewed availing the language editing service by LetPub, which is recommended by the American Society for Microbiology.

Line 54: 'living creatures?'

Modifications: Thank you for your suggestion. The phrase has been revised according to your suggestion (line 54 of the revised MS).

Line 61: occurring to

Modifications: Thank you for your suggestion. The expression has been revised according to your suggestion (line 61 of the revised MS).

Line 67-68: synthesized through 2-step

Modifications: Thank you for your suggestion. The phrase has been revised according to your suggestion (lines 67–68 of the revised MS).

Line 77: besides its role

Modifications: Thank you for your suggestion. The phrase has been revised according to your suggestion (line 77 of the revised MS).

Line 114: 'behaved'?

Modifications: Thank you for your suggestion. The expression has been revised according to your suggestion (line 114 of the revised MS).

Line 125-136: this section appears with weak consistency with other parts, and associated discussion may be necessary

Modifications: Thank you for your suggestion. The section has been revised according to your suggestion (lines 381–387 of the revised MS).

Line 195: what does 'slight' stand for? Is it of statistical significance or just a numerical decrease?

Modifications: Thank you for your suggestion. From the obtained data, we could see that deleting *cpxR* does not affect the expression of *acrA*, *acrB*, and *soxS* (Figure 4, Table S6). Though deleting *cpxR* does lead to decreased expression of *tolC* according to statistical analysis (0.02, $P < 0.05$), it is a less than two-times decrease comparing the wild type strain (0.63, $\Delta cpxR$ VS WT). The relevant content has been modified according to your suggestion (lines 199–201 of the revised MS).

October 30, 2021

Prof. Jiaoyu Deng
Key Laboratory of Special Pathogens and Biosafety, Wuhan Institute of Virology, Chinese Academy of Sciences
middle street of xiaohongshan, NO.44
Wuhan, Hubei 430071
China

Re: Spectrum00743-21R1 (Reducing the Periplasmic Glutathione Content Makes Escherichia coli Resistant to Trimethoprim and Other Antimicrobial Drugs)

Dear Prof. Jiaoyu Deng:

Your manuscript has been accepted, and I am forwarding it to the ASM Journals Department for publication. You will be notified when your proofs are ready to be viewed.

Sincerely,

Hui Wang
Editor, Microbiology Spectrum

Journals Department
TABLE S3: Accept
Supplemental file 1: Accept